# An Automatic Method for Assessing Spiking of Tibial Tubercles Associated with Knee Osteoarthritis

**DOI:** 10.3390/diagnostics12112603

**Published:** 2022-10-27

**Authors:** Anri Patron, Leevi Annala, Olli Lainiala, Juha Paloneva, Sami Äyrämö

**Affiliations:** 1Faculty of Information Technology, University of Jyväskylä, 40014 Jyväskylä, Finland; 2Department of Radiology, Tampere University Hospital, 33520 Tampere, Finland; 3Faculty of Medicine and Health Technologies, Tampere University, 33520 Tampere, Finland; 4Department of Surgery, Central Finland Healthcare District, 40620 Jyväskylä, Finland; 5Institute of Clinical Medicine, University of Eastern Finland, 70210 Kuopio, Finland

**Keywords:** knee joint, osteoarthritis, radiography, tibial spiking, convolutional neural networks

## Abstract

Efficient and scalable early diagnostic methods for knee osteoarthritis are desired due to the disease’s prevalence. The current automatic methods for detecting osteoarthritis using plain radiographs struggle to identify the subjects with early-stage disease. Tibial spiking has been hypothesized as a feature of early knee osteoarthritis. Previous research has demonstrated an association between knee osteoarthritis and tibial spiking, but the connection to the early-stage disease has not been investigated. We study tibial spiking as a feature of early knee osteoarthritis. Additionally, we develop a deep learning based model for detecting tibial spiking from plain radiographs. We collected and graded 913 knee radiographs for tibial spiking. We conducted two experiments: experiments A and B. In experiment A, we compared the subjects with and without tibial spiking using Mann-Whitney U-test. Experiment B consisted of developing and validating an interpretative deep learning based method for predicting tibial spiking. The subjects with tibial spiking had more severe Kellgren-Lawrence grade, medial joint space narrowing, and osteophyte score in the lateral tibial compartment. The developed method achieved an accuracy of 0.869. We find tibial spiking a promising feature in knee osteoarthritis diagnosis. Furthermore, the detection can be automatized.

## 1. Introduction

Knee osteoarthritis (OA) is a highly prevalent chronic joint disease and a prominent global cause of disability. In the Global Burden of Disease 2010 study, knee and hip OA was ranked the 11th most common global cause of disability [1]. As the prevalence of OA increases with age [2], due to population aging, the burden of OA is expected to rise. Early detection of OA is imperative for maximizing the efficacy of interventions, lowering the burden of the disease and the incidence of knee joint replacement surgery [3,4].

Plain radiography is a standard imaging modality for OA diagnosis and severity assessment. Radiographic signs of knee OA include joint space narrowing (JSN), formation of osteophytes, cysts, and subchondral sclerosis [5]. Plain radiography as an imaging modality is insensitive to early signs of knee OA, such as cartilage damage and minor osteophytes [6,7], which makes the radiographic diagnosis of early knee OA challenging.

The most common classification for radiological knee OA was described by Kellgren and Lawrence (KL) [8]. KL classification consists of ordinal grades from 0 to 4, where 0 stands for no signs of OA, and each subsequent grade signifies increasing OA severity. KL is a composite grading system, defining OA by the presence of JSN and osteophytes. The definitions of KL-grades for the knee joint are the following [8,9]:Grade 0: No radiological signs of OA.Grade 1: Doubtful JSN, possible osteophytic lipping.Grade 2: Definite osteophytes, possible JSN.Grade 3: Moderate multiple osteophytes, definite JSN, some sclerosis, possible deformity of bone ends.Grade 4: Large osteophytes, marked JSN, severe sclerosis, definite deformity of bone ends.

The KL system has been criticized for ambiguity, e.g., in cases where JSN is present, but osteophytes are not [10]. Osteoarthritis Research Society International (OARSI) developed another radiographic atlas for OA [11]. Unlike KL, OARSI contains individual grades for JSN and osteophytes on a scale of 0–3.

To increase the reliability and objectivity of OA assessment, fully automatic methods have been developed for assessing radiographic knee OA using plain radiographs. Oka et al. [12] developed an automatic method for quantifying features of OA, such as minimum joint space width and osteophyte area using filters and differentiation. Shamir et al. [13] used various handcrafted features extracted from the plain radiographs together with a weighted nearest neighbor classifier to predict the KL-grade. More recently, convolutional neural networks (CNN) [14] have achieved success in medical image classification tasks such as malignant skin lesion classification [15] or radiographic identifying of subjects with arthroplasty [16].

Using CNNs for knee OA severity assessment from plain radiographs was initially proposed by Antony et al. [17]. The recent state-of-the-art for predicting knee OA severity using deep learning has been reviewed by Yeoh et al. [18]. CNN-based methods have been reasonably successful in assessing severe KL-grades (i.e., grades 3 and 4), but for predicting grades marking early OA, the accuracy is notably lower [17,19,20]. The limitations of automation of early OA severity assessment using plain radiographs are likely multifaceted. Firstly radiographs do not allow for direct visualization of the cartilage. Furthermore, the current CNN-based methods are constrained by the KL system. As the current methods are trained using noisy KL scores as the ground truth, the resulting models thus capture the bias inherent to the KL classification. The results by Kim et al. [21] indicate that early OA severity assessment with CNN can be improved by providing the model with additional clinical information (e.g., age, sex, and body mass index (BMI)).

The limitations of early OA assessment could perhaps be alleviated further by considering additional radiographic features of OA not incorporated in the KL system, such as the spiking of tibial tubercles that has been hypothesized as a sign of early knee OA (hypothesis A). The spiking of tibial tubercles or tibial spiking refers to the tall and angular appearance of the tibial spines (see Figure 1). The first mention of hypothesis A, to our knowledge, is from a radiological textbook by Sutton [22]. However, the author provides no evidence for the said hypothesis.

The feature was later studied by Reiff et al. [23], who examined radiographs from fifty-five subjects with established knee OA and thirty-six controls. They found the lengthening and sharpening of the peaks of the tubercles associated with knee OA. Donnelly et al. [24] conducted a study with 950 subjects examining tibial spiking as a radiological feature of OA. They found the sharpening of tibial spines to correlate with OA status (defined by KL 2 or higher) and osteophyte scores. They, however, concluded that tibial spiking is not a reliable marker for knee OA in isolation due to a lack of clear independent association with knee pain [24].

Unluet al. [25] studied the association between tibial spiking and cartilage defects assessed via magnetic resonance imaging (MRI). The study involved seventy-six knees from forty-seven subjects and thirty-one knees from sixteen controls. The subjects with knee OA had significantly higher and sharper tibial spines than the controls. They observed a correlation between cartilage defects and medial tubercle height but not with lateral tubercle height [25]. Additionally, an association between the spiking of lateral tubercle and osteophyte formation in the tibial compartments was found [25]. The latest study by Hayeri et al. [26] considered tibial spiking from a paleopathological framework, where thirty-five tibial bone specimens were directly examined for signs of OA. The study found spiking of the lateral spine associated with osteophyte formation.

The previous research indicates that tibial spiking might be associated with knee OA and osteophyte formation [23,24,25,26], however, the feature might not be a reliable marker of knee OA in isolation [24]. Although osteophyte formation is one of the primary signs of radiographic OA, therefore tibial spiking might be a beneficial feature in assessing knee OA in cases where no evident osteophytes can be detected. With automatic methods, the cost of assessing radiographs for markers of OA is negligible and therefore provides a different value proposition compared to general clinical adoption. Provided that tibial spiking is identifiable by human experts, an automatic method can be developed, provided sufficient data (hypothesis B).

The aim of the present study was to evaluate the hypothesis on tibial spiking as an early sign of knee OA (hypothesis A). While the previous work on the subject indicates that tibial spiking might be a feature of knee OA [23,24,25,26], the research is still lacking. Especially whether tibial spiking is connected with the early knee OA is unclear. Furthermore, we examined the feasibility of identifying tibial spiking automatically by developing a method for assessing the feature from plain radiographs (hypothesis B), which has not been considered previously.

## 2. Materials and Methods

### 2.1. Radiographic Data

The present study utilized data from the Osteoarthritis Initiative (OAI) [27] and the Multicenter Osteoarthritis Study (MOST) [28]. OAI and MOST are longitudinal cohort studies of OA and include radiographs assessed for signs of OA at multiple time points. OAI dataset includes data from 4607 participants between ages 45–79 at baseline. MOST baseline dataset contains data from 3026 participants between ages 50–79.

We collected bilateral PA (posterior-anterior) fixed flexion knee radiographs with KL-grades 0–2 from OAI and MOST baseline datasets. The radiographs were selected randomly with an approximately equal number of samples from each KL-grade. We collected 722 radiographs from OAI and 191 radiographs from MOST, from which we used only the right knee. The collected knees with regard to the KL-grade can be seen in Table 1. Additionally, we collected assessments of radiographic knee OA, including KL-score, OARSI osteophyte and JSN scores, Western Ontario and McMaster Universities Osteoarthritis Index (WOMAC) knee pain score, and the subject BMI for each sample.

### 2.2. Data for Experiment A

As the tibial spiking was not assessed in OAI or MOST cohorts, we collected the assessments manually for all 913 knees. The assessments were performed by two physicians, a radiology resident with three years of experience in radiology (expert 1) and an experienced orthopedic surgeon (expert 2). Each knee was assessed by a single expert, blinded to the OA assessments and clinical details. Prior to grading the knees, the experts had a single session to establish a uniform view of the spiking criteria. Each spine (medial and lateral) was rated for spiking by subjective visual inspection of angulation, size, and other deformities. No angle or height measurements of the tubercles were performed in this study. The spines were graded on a binary scale of 0–1 with the possibility of giving an “unsure” rating. The unsure rating was warranted as the tibial spines might be occluded by the femur or otherwise difficult to judge, e.g., due to poor exposure. The spines were also rated unsure in borderline cases (i.e., doubtful spiking). We defined overall spiking (i.e., spiking on medial or lateral side) as
(1)spiking:=(lateralspiking=1)∨(medialspiking=1).

### 2.3. Reliability

We evaluated inter-rater reliability using a subset of 205 radiographs rated by both experts in separate sessions, blinded to the assessments made by the other party. Intra-rater reliability was assessed with a subset of the knees re-rated by the same expert, blinded to the previous rating. The duplicate radiographs used for evaluating reliability were mixed among the set of regular radiographs. Additionally, the experts were blinded to the existence of duplicate radiographs. Sample sizes for evaluating intra-rater reliability were 53 and 68 for experts 1 and 2, respectively. The inter- and intra-rater reliability were measured using Cohen’s κ (Kappa) [29]. For calculating κ scores we used implementation in Python (ver. 3.10.0) [30] library scikit-learn 1.0.1 [31].

The reliability analysis was performed for the original 3-way ratings (including the unsure ratings), binary (0–1) ratings where the unsure assessments were omitted, and overall spiking (defined in Equation (Equation 1)). The rating pairs were omitted if either contained an unsure rating. The sample sizes for assessing the intra-rater reliability of unsure omitted ratings were 41 and 42 for experts 1 and 2, respectively, and 124 for inter-rater reliability.

### 2.4. Experiment A

To evaluate the hypothesis on tibial spiking as a feature of early OA (hypothesis A), we conducted experiment A, where the differences were tested between groups with tibial spiking and a control group (i.e., subjects without tibial spiking). Given hypothesis A, we would expect the group with tibial spiking to have a higher KL-score, osteophyte-score, JSN-score, and knee pain. We combined the samples from OAI (722) and MOST (191), for 913 samples in experiment A.

We defined the inclusion criteria for the spiking group identically to overall spiking; see Equation (Equation 1). Consequently, the control group included all the samples with negative or unsure ratings. In total, 630 samples were assigned to the spiking group and 283 to the control group. For testing the group differences, we used two-tailed Mann-Whitney U-test [32]. We used the Scipy 1.7.3 [33] implementation for calculating the U-test values. To counteract the multiple comparisons problem, we applied the Bonferroni correction [34]. With significance levels α=(0.05,0.01,0.001) and the number of tests n=9, the corrected significance level $\alpha_{i}^{\prime }$ is
(2)$$\alpha_{i}^{\prime }=\frac{\alpha_{i}}{n}.$$

Some radiographs in the original OAI and MOST data were missing some OARSI assessments (JSN or osteophytes grade). We omitted the samples containing missing values for the calculations, which reduces the sample size for these variables. Sample sizes for the variables containing missing information are the following: BMI: spiking 629 and control 283, OARSI JSN variables: spiking 618 and control 281, and OARSI osteophyte variables: spiking 432 and control 145.

### 2.5. Data for Experiment B

We used 80% of OAI data (577 images) for model training and 20% (145 images) for model validation. All 191 images from MOST were used as the final test data. For a breakdown of each dataset split with regard to the tibial spiking rating, refer to Table 2. As the ground truth, we used the definition for overall spiking (see Equation (Equation 1)). We used the assessments from a single expert chosen randomly for the images graded by both experts.

The image data in the OAI and MOST datasets were stored as Digital Imaging and Communications in Medicine (DICOM) image format. We used pydicom 2.2.2 [35] for reading the DICOM pixel data. The original bilateral PA images were first localized to the region of interest (ROI), i.e., the right knee joint area. The images were localized by manually annotating a center point in the valley between medial and lateral tibial tubercles and calculating square ROI of size 300×300 from the center point. For annotating the ROI centers, we used labelme 5.0.1 [36]. Finally, we downsampled the ROI images to an input size of 224×224 by bilinear interpolation.

Following the localization, we inverted the pixel values of images with MONOCHROME1 photometric interpretation (i.e., we converted black-on-white images to white-on-black). We performed histogram equalization using OpenCV Python library [37] to improve the image contrast. All image sample pixel intensities were normalized by subtracting the mean and dividing by the standard deviation, which were calculated from the set of training samples.

We replaced the original training samples with augmented samples in a one-to-one fashion. The augmentations included flipping the images horizontally with a probability of 0.5 and performing random affine transformation (i.e., scaling, spatial translation, and rotation) to introduce variance among the training samples. The degree of rotation was sampled from a range of (–12, 12). The degree of spatial translation was sampled from (–11.2, 11.2) and (–2.24, 2.24) for the horizontal and vertical axis, respectively. The scale factor was sampled from (0.8, 1.2). The augmentations were resampled for each training iteration. For an illustration of the data processing pipeline, see Figure 2.

### 2.6. Experiment B

We developed and evaluated a model for identifying tibial spiking from plain radiographs to determine whether tibial spiking is detectable with automatic methods (hypothesis B). For the classification model, we fine-tuned (i.e., domain adapted) a CNN model ResNeXt-50-32x4d introduced by Xie et al. [38] pre-trained on around 1.2 million color images from ImageNet [39] challenge [40] for assessing tibial spiking. The CNN model implementation from Torchvision [41] version 0.12.0 was used. We modified the model by replacing the dense layer with two dense units, followed by softmax. We, therefore, initialized the weights for all other layers pre-trained on ImageNet. The motivation for the procedure is to improve the performance of CNN by utilizing the features learned from another dataset [42].

The CNN model architecture is detailed in Table 3. The bottleneck blocks, i.e., BN1–4 in Table 3 featured a shortcut connections [38] similar to He et al. [43]. The shortcut connection for a block B is defined as
(3)y=B(x)+x,
where *y* is the output and *x* is the input. Note that the dimensions of the block B and the identity *x* must be equal in Equation (Equation 3). When this is not the case, a linear projection $W_{d}x$ is used to match the dimensions before adding the identity [43]. We used the standard cross-entropy as the loss function, optimized with Adam [44] with following parameters $\alpha =0.0002,\;\beta_{1}=0.9,\;\beta_{2}=0.999,\;\epsilon =10^{-8}$. PyTorch 1.11.0 [41] was used as the model training and testing framework.

We performed a grid search to determine suitable hyperparameters for fine-tuning the classification model. As the model selection criteria, we used validation accuracy. The parameter space used in the grid search can be seen in Table 4. The best-performing model was trained for ten epochs with a batch-size four and a learning rate (α) of 0.0002. The learning rate was decayed every four epochs by a factor of 0.115.

We used Gradient-weighted Class Activation Mapping (Grad-CAM) [45] implementation TorchCAM (ver. 0.3.1a0) [46] for visualizing the model predictions. Grad-CAM provides visual explainability by highlighting the regions from the input image strongly influencing the output [45]. The heatmaps generated by Grad-CAM provide transparency into the model prediction-making and enable the developers to determine how the model is able to discern the classes or fails to do so. For the users, the visual explanations enable the building of trust in the classifier system. The study workflow and the methodology are summarized in Figure 3. The analysis code and the trained model for detecting tibial spiking have been made available on GitHub: https://github.com/AI-hub-keskisuomi/AI_hub_keskisuomi/tree/main/WP3_knee_osteoarthritis/tibial_spiking_grading (accessed on 4 October 2022).

## 3. Results and Discussion

### 3.1. Reliability

The reproducibility of tibial spiking grading is detailed in Table 5. According to a frequently used scale reported by Landis and Koch [47] for interpreting κ values, the range 0.21–0.40 represents fair agreement, 0.41–0.60 moderate agreement, 0.61–80 substantial agreement, and 0.81–1.00 almost perfect agreement. Albeit, the ranges are arbitrary according to Landis and Koch [47] but are often used to discuss reliability analysis results.

Inter-rater reliability for the lateral spiking was in the moderate range, similar to the KL-grade reliability evaluated in previous studies [48,49]. However, the inter-rater reliability of the medial side assessments was only fair. The medial spiking ratings contained more unsure ratings, indicating that the medial spines were more challenging to assess. The proximate cause for the discrepancy can only be conjectured. Nevertheless, the medial tibial tubercles are more prominent and thus are more likely to be occluded by the femur. After omitting the unsure ratings, the inter-rater reliability was comparable to the KL grading reliability.

The intra-rater reliability of both experts was moderate to substantial for the 3-way ratings. The intra-rater reliability of expert 2 was lower for the medial side. However, expert 1 was equally consistent in assessing the lateral and the medial sides. The disparity in the ratings of expert 2 seems to be explainable by the unsure ratings. The ratings for the knees graded twice of expert 2 contained 38% unsure ratings, while assessments of expert 1 contained 23% unsure ratings. The intra-rater reliability of expert 2 was near-perfect after omitting the unsure ratings.

Overall, the intra-rater reliability for assessing tibial spiking was comparable to KL grading (0.50 weighted κ with 95% confidence interval (CI) of (0.25–0.75)) [48]. After removing the unsure ratings, the intra-rater reliability of expert 2 exceeded the κ reported by Gossec et al. [48]. Although, intra-rater reliability of assessing tibial spiking seems less reliable than KL grading when compared against the KL intra-rater reliability of 0.97 weighted κ with 95% CI of (0.92–1.0) reported by Culvenor et al. [49]. After omitting the unsure ratings, the intra-rater reliability of expert 2 was comparable to the weighted κ obtained by Culvenor et al. [49].

### 3.2. Experiment A

The sample means for spiking and control groups jointly with U-test significance levels are detailed in Table 6. After applying the Bonferroni correction, the significant variables below p<0.001 were KL-grade and BMI, and the variables below p<0.01 were OARSI Medial JSN and tibia lateral osteophytes score.

We found significant group differences supporting hypothesis A. Higher KL, OARSI medial JSN and lateral tibia osteophyte grades in the spiking group support the previously reported evidence for the association between tibial spiking and knee OA [23,26]. However, there was no difference in the less prominent [5] lateral JSN. Considering the spiking group’s higher mean KL-grade, the higher BMI in the spiking group is consistent with the literature on the risk factors of knee OA [50]. Our results partly confirm the association between tibial spiking and osteophytes reported previously [24,25,26]. However, we could not confirm an association between tibial spiking and knee pain.

### 3.3. Experiment B

We evaluated the top model from the grid search using 191 samples from the MOST dataset. The model produced an accuracy of 0.869 with sensitivity of 0.909 and specificity of 0.750. More details on the model performance can be seen in Table 7. Confusion matrix for the test data is presented in Figure 4.

The developed model obtained lower specificity than sensitivity, meaning the model suffers to a greater extent from type I error (i.e., the model predicted spiking when none was present), which might reflect the class imbalance in the training samples. Additionally, due to how the ground truth was constructed, the no-spiking class might have contained more borderline cases (a subset of the unsure ratings could have been regarded as doubtful spiking). In future studies, more data should be accumulated to address the asymmetry.

Like knee OA severity grading, automating tibial spiking detection lacks the “ideal” ground truth (i.e., 100% reliable labels). Consequently, the models derived will be constrained by the quality of data available. The ground truth’s inter-rater reliability (κ) was 0.48, i.e., moderate. Currently, a universally agreed-upon atlas for assessing tibial spiking does not exist. The lack of shared criteria for grading tibial spiking casts doubt on the generalizability of the model developed in the present work, as different experts might have divergent views on how the spiking of tibial tubercles manifests in the radiographs. Nevertheless, the results of experiment A indicate that the spiking the model was trained to detect is associated with the other signs of knee OA.

It should be emphasized that calling the feature tibial “spiking” is somewhat imprecise in the present work, as the feature was graded based not only on the angulation of the spines but also on the length and bony growth on the spine peaks. Therefore, the feature might be more explicitly considered as “osteophytic” abnormalities in the tibial tubercles. Alexander [51] speculated that tibial spiking might be a type of osteophyte formation. Our results give some support for the theory. However, more research on the topic is required as the exact mechanism behind tibial spiking is unknown.

We visualized the model predictions for the validation data using Grad-CAM. By visualizing the failed predictions of the model, we can gain information on the reasons behind the failures, e.g., in Figure 5a, the network concentrates on the narrow joint space instead of on the tubercles. However, in Figure 5b, the model concentrates on the medial tibial tubercle predicting spiking while the ground truth was non-spiking. The heatmap indicates a strong influence of the medial tubercle; the assessments could be re-evaluated in cases where the model has an apparent disagreement with the expert. The model used in this manner can provide a second opinion for a physician.

The successful predictions can be visualized to identify how the model is able to discern the spiking samples from the non-spiking samples. In Figure 6a, the model has concentrated on the vicinity of the lateral tubercle. In Figure 6b, the model could not identify any spiking features and consequently predicted non-spiking.

## 4. Conclusions

Our results indicate an association between the spiking of tibial tubercles and early knee OA. Adopting spiking of tibial tubercles as additional information in the diagnosis of knee OA seems promising. Although, additional research on the characteristics of tibial spiking and guidelines for assessing the feature is needed. The model developed for automatically identifying tibial spiking was able to generalize despite the modest number of training samples. The analysis using Grad-CAM revealed that the developed method is somewhat reliant on the JSN, which could lead to misclassifications. In the future, more data on tibial spiking should be acquired to develop better tools for healthcare.

## Figures and Tables

**Figure 1 diagnostics-12-02603-f001:**
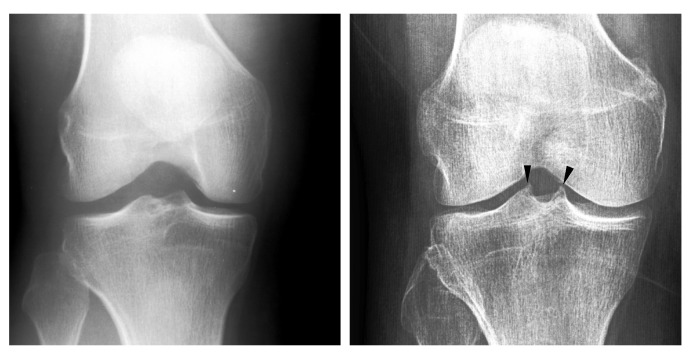
Two knee radiographs rated for spiking of tibial tubercles are compared. The tubercles rated as spiking are indicated by arrowheads.

**Figure 2 diagnostics-12-02603-f002:**
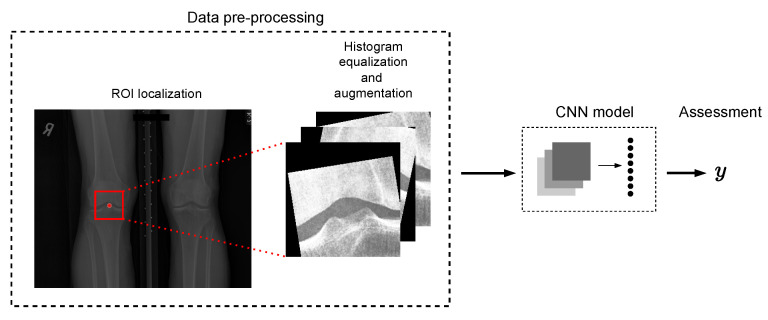
Tibial spiking assessment pipeline: the bilateral PA view radiographs are localized to the ROI, histogram equalization, augmentations (when applicable), and normalization are performed before feeding the data to the model.

**Figure 3 diagnostics-12-02603-f003:**
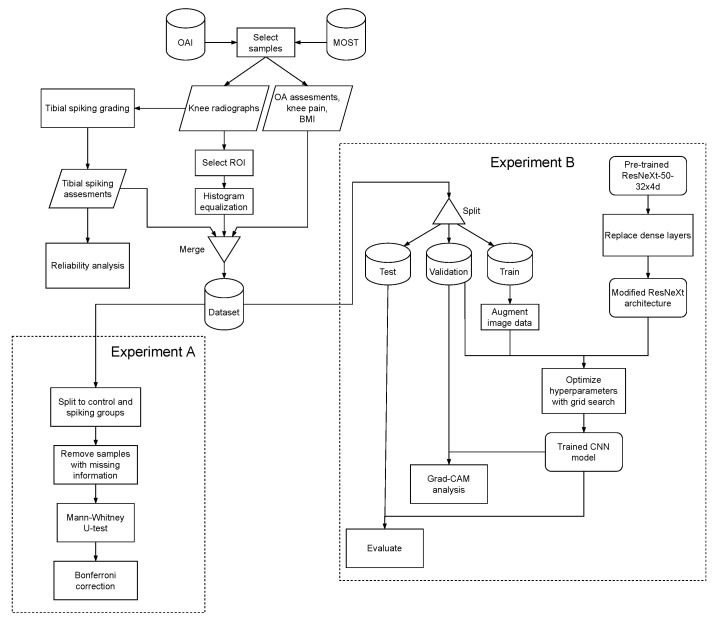
Flowchart of the study methodology. The processes or methods are denoted with rectangles, datasets with cylinders, data with parallelograms, and the models with rectangles with rounded corners.

**Figure 4 diagnostics-12-02603-f004:**
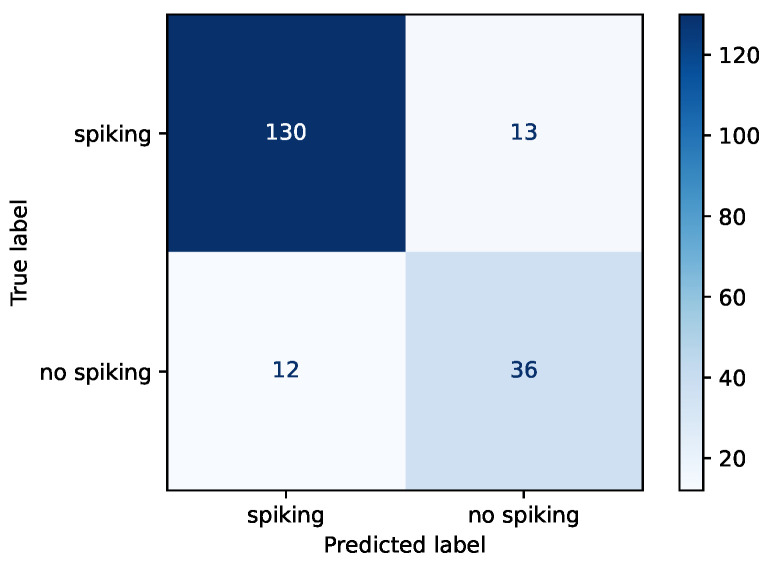
Confusion matrix for the test dataset.

**Figure 5 diagnostics-12-02603-f005:**
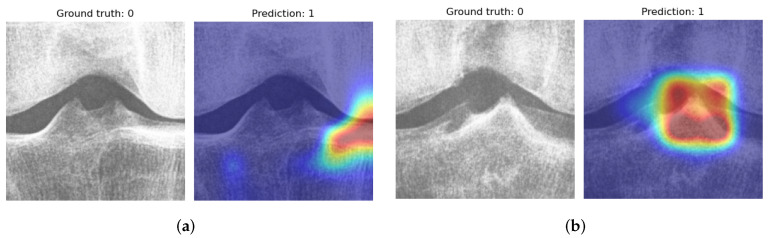
Grad-CAM visualizations for incorrect predictions. In subfigure (**a**), the model rates the knee spiking based on the narrow appearance of the medial joint space, which indicates that the model has learned the association between tibial spiking and medial JSN. In subfigure (**b**), the model makes a spiking assessment based on the medial tubercle.

**Figure 6 diagnostics-12-02603-f006:**
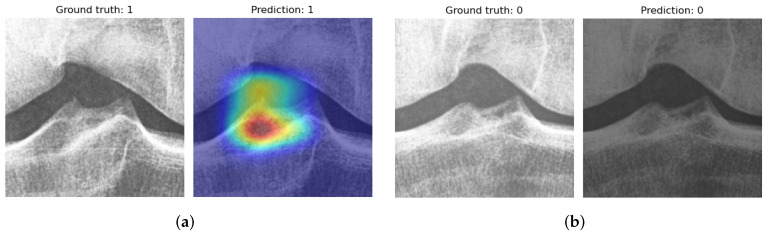
Grad-CAM visualizations for correct predictions. In subfigure (**a**), the lateral tubercle had the highest contribution to the prediction. In subfigure (**b**), no influential region was found from a non-spiking sample, as can be observed from the blank heatmap.

**Table 1 diagnostics-12-02603-t001:** Collected knees with regard to KL-grade.

	KL0	KL1	KL2
OAI	243	248	231
MOST	65	61	65

**Table 2 diagnostics-12-02603-t002:** Tibial spiking data frequency tables for medial, lateral, and overall spiking (medial or lateral; OR). Zero indicates the absence of spiking, while one indicates the presence of spiking. The question mark indicates the unsure rating.

	Medial			Lateral			OR	
	**0**	**1**	**?**	**0**	**1**	**?**	**0**	**1**
Train	216	281	80	253	279	45	188	389
Validation	57	71	17	53	77	15	47	98
Test	66	108	17	59	116	16	48	143
Total	339	460	114	365	472	76	283	630

**Table 3 diagnostics-12-02603-t003:** The architecture of the used model consists of sequential bottleneck (BN) blocks after the initial convolution and max pooling. The convolution layer parameters are presented in the order of kernel size, and the number of kernels and *C* denotes the number of grouped convolutions. A bracket followed by ×k indicates the block is repeated *k* times. The dense layer has an input size of 2048 and an output size of two. The spatial output size of the block is presented in the middle column.

Block	Output Size	Architecture	
Conv1	112×112	7×7, 64, stride 2	
Pool	56×56	3×3 max pool, stride 2	
BN1	56×56	1×1, 1283×3, 128, C=321×1, 256	}×3
BN2	28×28	1×1, 2563×3, 256, C=321×1, 512	}×4
BN3	14×14	1×1, 5123×3, 512, C=321×1, 1024	}×6
BN4	7×7	1×1, 10243×3 1024, C=321×1, 2048	}×3
	1×1	global average pool	
Dense		2048, 2, softmax	

**Table 4 diagnostics-12-02603-t004:** Grid search parameter space. Step-size defines the interval for decaying the learning rate specified by gamma.

	Values
Epochs	1,2,3,…,23,24,25
Batch-size	4,5,6
Learning-rate	$7\times 10^{-3},\;8\times 10^{-3},\;9\times 10^{-3},\;10^{-4},\;2\times 10^{-4},\;3\times 10^{-4},\;4\times 10^{-4}$
Step-size	4,5,6
Gamma	0.115,0.12,0.125

**Table 5 diagnostics-12-02603-t005:** Intra and inter-rater reliability (κ) for 3-way ratings, binary ratings with unsure omitted (denoted with o), and overall spiking (denoted with OR) with 95% confidence interval (CI).

	Intra-Rater Reliability (Expert 1)	Intra-Rater Reliability (Expert 2)	Inter-Rater Reliability
Medial	0.61 (0.58–0.64)	0.52 (0.50–0.54)	0.34 (0.33–0.35)
Medial (o)	0.78 (0.75–0.82)	0.94 (0.92–0.96)	0.59 (0.58–0.61)
Lateral	0.59 (0.56–0.62)	0.75 (0.73–0.76)	0.55 (0.55–0.56)
Lateral (o)	0.71 (0.67–0.74)	1.00 (1.00–1.00)	0.75 (0.74–0.76)
OR	0.53 (0.50–0.57)	0.69 (0.67–0.72)	0.48 (0.47–0.49)

**Table 6 diagnostics-12-02603-t006:** Mean values for spiking and control groups with U-test significance levels (without Bonferroni correction * p<0.05; ** p<0.01; *** p<0.001).

	Spiking	Control
KL-grade ***	1.11	0.70
WOMAC knee pain *	2.14	1.62
BMI ***	29.09	27.46
Medial JSN ***	0.38	0.25
Lateral JSN	0.05	0.04
Tibia medial osteophytes **	0.59	0.41
Tibia lateral osteophytes ***	0.40	0.21
Femur medial osteophytes **	0.48	0.27
Femur lateral osteophytes **	0.41	0.22

**Table 7 diagnostics-12-02603-t007:** Classifier performance metrics.

	Accuracy	Loss	Sensitivity	Specificity	Precision
Train	0.872	0.300	0.882	0.851	0.925
Validation	0.869	0.399	0.929	0.745	0.883
Test	0.869	0.314	0.909	0.750	0.915

## Data Availability

Data available in a publicly accessible repository that does not issue DOIs Publicly available datasets were analyzed in this study. OAI data can be found here: https://nda.nih.gov/oai/ (accessed on 4 October 2022). MOST data was available at: https://most.ucsf.edu (currently unavailable as of 4 October 2022).

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
