# Peer review of "An Automatic Method for Assessing Spiking of Tibial Tubercles Associated with Knee Osteoarthritis"

_diagnostics, 2022, doi:10.3390/diagnostics12112603_

Round 1

Reviewer 1 Report

Reviewer comments

Title Automatic Method for Assessing Spiking of Tibial Tubercles Associated with Knee Osteoarthritis

The topic is interesting and well written and illustrated. However, some points need to be addressed

Line

Manuscript

Comment

72

In the present study, we conducted two experiments corresponding to 72

hypotheses A and B.

The last part of the introduction section usually includes the research gap and the aim of the study

It is not appropriate to write the summary of the study here

83

The present study utilizes data

It is better to write the materials and methods section in the past tense

Radiographic evaluation  

More details is needed about this point

126

It will be better if the author add a flow chart for the study

138

labelme 5.0.1 [? ].

Is this a missing reference

142

library [? ]

Is this a missing reference

The manuscript needs more details on tibial spiking as an early presentation of osteoarthritis and the studies covered this point

Reviewer 2 Report

The authors of this work investigate tibial spiking as a characteristic of early knee osteoarthritis. They also created a deep learning-based model for identifying tibial spiking on plain radiographs.

The authors had described the methodology in a good manner but the details about the proposed deep learning based model for detecting tibial spiking from plain radiographs is not sufficient. Hence more detailed explanation is required about the proposed model. If it is sufficient, in the next submission, then only the paper can be considered for further procedures.

Graphical or tabular comparison of results obtained with other similar works in the literature is missing.

Round 2

Reviewer 1 Report

The manuscript has been improved